# Effects of Lingual and Palatal Site Toothbrushing on Periodontal Disease in the Elderly: A Cross-Sectional Study

**DOI:** 10.3390/ijerph18105067

**Published:** 2021-05-11

**Authors:** Byung-Ik Yang, Ji-A Park, Jae-Young Lee, Bo-Hyoung Jin

**Affiliations:** 1Department of Preventive Dentistry & Public Oral Health, School of Dentistry, Seoul National University, Seoul 08826, Korea; biolaw@snu.ac.kr (B.-I.Y.); narcist0@snu.ac.kr (J.-A.P.); 2Dental Research Institute, Seoul National University, Seoul 08826, Korea; 3Department of Dental Hygiene, College of Health Science, Dankook University, Chungnam 31116, Korea

**Keywords:** plaque control, bleeding on probing, periodontitis

## Abstract

(1) *Background*: To analyze the association between periodontal health status and daily oral health activities including lingual and palatal site toothbrushing. (2) *Methods*: One hundred and fifty Korean elderly people aged >65 years participated in the study. Clinical examination regarding oral health status, including periodontal health status, was evaluated, and data on the oral health activities, socio-demographic factors, and systemic health-related factors were obtained using a questionnaire. Statistical analyses assessed the differences of periodontal health status according to daily oral health activities, including lingual and palatal site toothbrushing. (3) *Results*: Oral health activities including lingual and palatal site toothbrushing, frequency of toothbrushing, use of an interdental toothbrush, dietary patterns, and activity dependence correlated with bleeding on probing (BOP) and periodontitis. After adjusting for covariates, the prevalence of periodontitis was lower in the group where the inner surfaces of the teeth were partially or wholly cleaned than in the group without such cleaning. High BOP was significantly associated with the brushing of the inner surfaces of teeth. (4) *Conclusions*: Lingual and palatal site toothbrushing was associated with good periodontal health status in the elderly; the importance of brushing the inner surfaces of teeth should be emphasized for them and their caregivers.

## 1. Introduction

One of the most significant issues in modern societies is aging. The proportion of the population aged >60 years is predicted to have nearly doubled from 12% to 22% worldwide [1]. The proportion of the elderly aged >65 years in South Korea was 14.9% in 2019 and is expected to reach 39.8% by 2050; the aging population has increased more rapidly than in any other country [2].

As the population ages, chronic diseases, such as diabetes, cardiovascular diseases, and degenerative diseases, also rise. Moreover, oral diseases, such as periodontitis or masticatory discomfort, have also increased. According to the latest Korean statistics, the prevalence of periodontal diseases among the population aged >65 years old was in the range of 40% and the proportion of oral function restriction and subjective chewing discomfort was 40.7% and 38.1%, respectively [3].

Maintaining good oral health and its function is essential for the vigor and systemic health of the elderly and for proper nutritional intake and digestion. Besides, oral diseases, especially periodontal diseases, are inter-related with systemic conditions, such as diabetes and cardiovascular health problems, as reported in previous studies [4,5,6]. Therefore, periodontal health management is critical in the elderly from the perspective of various chronic diseases, considering that oral and noncommunicable diseases are affected by common factors, such as smoking, alcohol, and obesity, oral microbial dysbiosis and the inflammatory process leading to microbial diseases is also a possible contributing factor [7,8,9]. Plaque control is the first step in periodontal healthcare and positively affects the elderly. However, with advancing age, oral health becomes one of the most neglected health domains, primarily because of social perceptions that undermine the importance of oral health or/and the focus on other severe diseases [10].

In addition, weakened general muscle and grip strength make maintenance of oral hygiene difficult [11]. The use of oral care products by older adults is lower than that of young adults, as is the frequency of toothbrushing, which is the primary step for removing the dental biofilm [12]. As a result, older people are susceptible to oral diseases.

Meanwhile, excellent oral cleanliness requires brushing the entire tooth surface evenly [13]. The tooth’s inner (lingual or palatal) surface is a difficult site to brush, with frequent accumulation of dental plaque and calculus, which causes gingival recession [14]. The lingual surface is a far better measure of periodontal health status and morbidity than the buccal surface [15]. It was reported that the ratio and time spent on cleansing palatal and lingual surfaces were significantly less than that for buccal or labial surfaces in children or young adults [16]. Also, the neglect of cleaning the inner surface was similar for both children and young adults, regardless of whether they did their best or brushed habitually [17]. Therefore, for the elderly with unfavorable conditions for oral hygiene care (e.g., poor mobility, bed-ridden state), brushing the inner surface may be more neglected or not efficiently done.

Daily oral health care activities were highly effective in maintaining a healthy oral status and mitigating against periodontal health inequality in the elderly [18]. Assessment of the relationship between periodontal status and various oral health behaviors (including cleaning the inner teeth surfaces) and the elderly’s oral health behaviors will help improve the efficiency of oral care activities and alleviate oral inequality. This information will also be necessary for the development of appropriate oral health education for the caregivers of the elderly.

Therefore, this study aimed to explore oral hygiene behaviors (instructions) that should be focused on for plaque control, which is the first step in preventing periodontal inflammation or periodontal diseases, as evaluated by the association between periodontal health status and oral health activities. In particular, this study hypothesized that among several oral health activities, palatal and lingual toothbrushing correlated with periodontal disease. The results would provide baseline data to direct oral health education for the elderly and their caregivers.

## 2. Materials and Methods

### 2.1. Study Design

This cross-sectional study was conducted from October 2019 to March 2020, targeting elderly participants aged >65 years. Written informed consent was obtained from each participant. Trained researchers conducted a questionnaire via face-to-face interviews with elderly participants for approximately 20 min. The questionnaire contents evaluated the socio-economic factors, general health status, and oral health activities of the elderly group, followed by a direct investigation of the participants’ oral health status by one dentist.

### 2.2. Populations

In this study, 150 elders were selected as study participants from Seoul, Korea. These elderly either resided in or used elderly welfare (or care) facilities that cooperate with community health centers. The population aged >65 years who agreed to participate in the oral health survey and questionnaire for oral health activities were surveyed. Participants with difficulties completing the questionnaire or with challenges in evaluating their oral health condition by the dentist were excluded from the study. The sample size was estimated based on the latest national oral health survey performed in 2017 [6], when the prevalence of periodontitis of the elderly (over 65-year-olds) in Korea was 47.6%. We set the 95% confidence interval at 10% and assumed a dropout rate of the questionnaire response at approximately 30%. Accordingly, at least 150 elderly participants had to be recruited.

### 2.3. Clinical Examinations

Examinations were performed using a portable dental unit (portable unit chair and compressor, Daeyang dentech, Seoul, Korea) equipped with a dental light, a dental mirror, and a periodontal probe. Oral health examination was conducted to assess the oral health status by trained dentists using the World Health Organization (WHO) oral health survey protocol [19]. A whole-mouth periodontal examination was conducted by the dentist using a periodontal probe (Hu-Friedy PCP-UNC 15, Hu-Friedy, Chicago, IL, USA). The periodontal pocket depth was measured at complete sites for each tooth (mesio-buccal, mid-buccal, disto-buccal, mesio-lingual, mid-lingual, and disto-lingual) as the distance from the gingival margin to the bottom of the sulcus. The periodontal status of the participants under the Community Periodontal Index (CPI) was classified as 0 for healthy, 1 (gingivitis with bleeding on probing) to 2 (presence of calculus) for gingivitis, and 3 (periodontal pocket depth of 3.5 mm or more) to 4 (periodontal pocket depth 5.5 mm or more) for periodontitis. In this study, the periodontal health status was classified as high CPI in cases with CPI 3–4. The bleeding on probing (BOP) was measured at the margin of the gums surrounding each tooth, divided into the central, lingual, and mesial sites. Based on the observations of presence of bleeding at these sites, the arithmetic mean of BOP was calculated. The BOP state was divided into two groups (low and high) based on the median score. The number of remaining teeth were measured.

### 2.4. Oral Health Activity

With the aid of a face-to-face questionnaire, various behaviors related to oral health and hygiene care were assessed. These included the need for help during toothbrushing, daily frequency of toothbrushing, use of an interdental brush, and inclusion of lingual and palatal surfaces in toothbrushing. Regarding the dietary factors, the questionnaire comprised questions assessing fresh seasonal vegetable intake as oral healthy foods.

These variables used as explanatory or confounding variables in this study were selected from the results of a previous Delphi investigation by experts on oral health activities (unpublished data) and based on other related studies [20,21,22]. They reported these oral health activities to be important determinants of periodontal health or plaque control.

### 2.5. Assessment of Confounders

Socio-demographic factors considered to be confounders in the present study included age, sex, education status, and household income. The income was categorized in four quartiles. Other variables considered included the participants’ dependence for activities of daily living (ADL) [23], smoking habits, and history of systemic diseases, including diabetes and cardiovascular diseases. This information was collected by direct questioning of the study participants.

### 2.6. Statistical Analysis

Participants’ characteristics (including socio-demographic factors, systemic health-related factors, oral health activities, and oral health status), characterized according to the periodontal health status including periodontitis (CPI ≥ 3 or CPI < 3) and BOP rate (high or low), were described using frequency distributions for categorical variables and mean values with standard deviations for continuous variables. The differences between categorical variables and normally distributed continuous variables were compared using Chi-square tests and t-tests. Multivariate logistic regression analyses were performed to quantify the relationship between periodontal health status and brushing of the inner surfaces of the teeth for oral health activities. The odds ratios (ORs) and 95% confidence intervals (CIs) of periodontitis and BOP rate according to the performance of lingual and palatal toothbrushing were calculated and adjusted for confounding variables. SPSS statistics program (IBM Corp. Released 2017. IBM SPSS Statistics for Windows, Version 25.0. Armonk, NY, USA) was used for the analyses.

## 3. Results

The characteristics of the study participants, according to the periodontal health status, are summarized in Table 1.

Of the 150 participants, 56 (37.3%) had a CPI ≥ 3, and the median of the participants’ BOP rate was 35% (interquartile range 15–66%), and 73 participants (48.7%) with a higher BOP rate than the median were considered as the ‘high BOP group’ (Table 1).

In the univariate analyses, the group with periodontitis had lower income, lower education level, and a higher dependence on daily living activities. Based on the oral health activities, they had a lower intake of fresh vegetables, higher need for assistance during toothbrushing, lower frequency of toothbrushing and use of interdental brushes, and lower performance of lingual and palatal site toothbrushing. Also, the differences between the high and low BOP groups had similar tendencies, like having a lower intake fresh vegetables and being highly dependent on toothbrushing, having lower frequency of toothbrushing, using fewer interdental brushes, and not brushing lingual and palatal sites in the high BOP group. Additionally, the high BOP group showed a higher DMFT and a lower number of remaining teeth than the low BOP group. The group with periodontitis had a higher BOP rate on average than did the group without periodontitis.

Periodontitis and BOP correlated with several oral health activities, such as fresh seasonal vegetable ingestion, help with toothbrushing, frequency of toothbrushing, interdental brush usage, and lingual and palatal site toothbrushing (*p*-value < 0.000 to *p*-value = 0.054, Table 1).

In preliminary regression analyses, the toothbrushing performance for the inner surfaces most strongly correlated with periodontal health status. Logistic regression analyses adjustments for confounding factors were done to assess the toothbrushing performance of the inner surfaces among the above-mentioned oral health activities. The significant associations between oral health behavior and periodontitis or high BOP are presented in Table 2.

Reference value is that brushing of the inner surfaces of teeth was not performed.

Model 1: Socio-demographic status (sex, age, income, education level), adjusted; Model 2: Socio-demographic status, systemic health status (systemic diseases, smoking), adjusted; Model 3: Socio-demographic status, systemic health status, other oral health status (remaining teeth), adjusted; Model 4: Socio-demographic status, systemic health status, other oral health status, other oral health activities (intake of fresh seasonal vegetables, need for help in toothbrushing, daily frequency of tooth brushing, use of interdental brush), adjusted.

The regressions from Model 1 to Model 4 were adjusted for socio-demographic factors, systemic conditions, oral health status except for periodontal status, and other oral health behaviors in order. The following model is one in which the new covariates were accumulated with the covariates of the previous model. The need for assistance in toothbrushing was excluded from the covariate; it was confirmed that it acted as a mediator variable when the ADL affected periodontal health status (data not shown).

The results from the multivariate logistic regression analysis revealed that even after correcting the various confounding factors, lingual and palatal site toothbrushing was significantly associated with periodontal health. The odds of periodontitis prevalence or high BOP in the group partially or wholly performing inner surface toothbrushing were significantly lower than the group not performing inner surface toothbrushing in all the adjusted models (adjusted ORs [CIs]: 0.067 [0.018–0.183] in Model 3 to 0.148 [0.064–0.345] in Model 2). Similarly, the performance of lingual and palatal site toothbrushing was also significantly associated with the BOP rate.

## 4. Discussion

This study explored the oral health activities that should focus on plaque control for preventing periodontal inflammation or periodontal diseases, as evaluated by the association between periodontal health status and oral health activities. The results provide data for directing oral health education for the elderly and their caregivers.

This cross-sectional study demonstrated that the lingual and palatal sites for toothbrushing are associated with periodontitis in elderly Koreans. Although dental technology developments have reduced the rate of oral health deterioration, individuals should continue to care for and manage their oral health before treatment [24,25,26]. Poor oral health could result in exacerbations of systemic diseases and significantly impact involuntary weight loss and aging. Therefore, maintaining and managing oral health among the elderly is essential. In this study, we confirmed various health conditions and activities, such as information on the oral health condition of the elderly, oral health activity, and general health conditions associated with oral health behavior. The use of lingual and palatal site toothbrushing was directly associated with periodontal health (Table 1).

According to previous research, greater oral health awareness is associated with superior oral health care behaviors [27,28]. This study shows that some elderly individuals may focus on brushing the buccal surfaces, which other people can see, with only 56.7% of participants partially or wholly brushing the tooth’s lingual surface (Table 1). The percentage of participants who performed lingual site toothbrushing may indicate that lingual toothbrushing is neglected when individuals require assistance from others, possibly due to a lack of cooperation from the elderly regarding oral hygiene management [29]. However, previous studies in Korea have reported a dearth in education on the areas of brushing in the oral cavity for older people who are not cooperative in oral hygiene management [30]. As a result, it is easy to neglect the care of the teeth’s lingual surface, including the lower anterior teeth, where food residue underflow and supragingival calculus are likely to accumulate due to the disappearance of oral muscle function [31]. Thus, oral health care practices to manage these sites are essential and oral health care products can reduce the risk of periodontal disease.

Other oral health activities, such as fresh seasonal vegetable ingestion, help with toothbrushing, daily frequency of tooth brushing, and use of interdental brush, were also significantly associated with a low prevalence of periodontitis and gingival bleeding in the preliminary regression analysis. We focused on the daily oral health activity one could practice but easily misses—lingual and palatal site toothbrushing. In addition, among the oral health activities, such as fresh vegetables intake, need for the toothbrushing the number of daily toothbrushing, use of the interdental brush, lingual and palatal site toothbrushing, it was found that the statistical explanatory power of the outcome variable was the largest.

Elderly, aging individuals who have low ADL or need help with brushing have higher levels of halitosis than those without any of these factors [32]. Al-Zahrani et al. [33] reported a relationship between physical activity and periodontitis prevalence. A national survey in America reported that a high level of physical activity prevented periodontitis. It has been found that physical health activities maintain a healthy periodontal state by creating an antibody of bacteria that makes the periodontal state healthy [34]. In addition, it has been reported that periodontal disease often occurs in elderly individuals with poor health activities and conditions [35]. However, there is an association between the need for assistance in toothbrushing-mediated ADL and periodontal health status. It has been reported that in the case of such discomfort, toothbrushing and dental plaque management through expert opinion can be of great help in improving oral health and quality of life [10]. Therefore, ADL was excluded in the covariate-adjusted models.

Some studies have reported that the frequency of cleansing palatal and lingual surfaces was significantly lesser than that of buccal or labial surfaces, even in children or young adults, and the negligence associated with cleaning the inner surface was similar to whether they did their best or brushed habitually [36,37]. This situation was more evident in the elderly, with unfavorable conditions for oral hygiene care associated with lower ADL. In addition, these patients have a high degree of dependence on a caregiver and need help when toothbrushing. In order to brush the lingual site more efficiently, it is recommended to use a toothbrush with a small head or an electric toothbrush to more effectively remove tooth surface bacteria [38].

A common site of dental plaque in adults and the elderly is the lingual surface of the lower anterior tooth. Thus, it is crucial to brush the inner surface due to frequent periodontal disease caused by calculus on the inner surface of the teeth [39]. Further, our study suggests that individuals lacking proper oral health education tend not to clean the inner surface of their teeth. Thus, it is important to position the bristles vertically toward the tooth surface for effective plaque removal. In addition, knowledge related to oral health is necessary to perform effective toothbrushing [40]. To obtain proper evidence for brushing education, high-quality research is required to provide more definitive guidelines on oral health promotion practices for the elderly [41].

In this study, following correction for sociological factors, medial tooth brushing strongly influenced periodontal disease in the elderly (8.98 times higher than in participants without medial tooth brushing). Further, we elucidated that lingual toothbrushing execution is not performed in those dependent on caregivers for ADL (Table 1). Recognition of and emphasis on the importance of internal tooth brushing when help is required are crucial. In particular, the capacity for ADL was reduced in some elderly; this group required help with toothbrushing. Therefore, recognition of the importance of brushing the tooth’s inner surface by families and caregivers who provide assistance in the oral hygiene care of the elderly is needed [42].

### 4.1. Strengths of the Study

The present study considered various health status criteria, including oral health status in the elderly measured via health measurement devices. The study is also supported by related research on toothbrushing as a well-established risk factor for periodontal disease. Although the exact effects of lingual site brushing are not understood, it should be considered a significant contributor to periodontal disease.

### 4.2. Limitations of the Study

This study is a cross-sectional study, reflecting the influencing factors according to the characteristics of the individual or group selected as a sample. Therefore, it is thought that it is necessary to reinforce the results through clinical studies, such as split mouth design, in the future. Further studies are needed to elucidate causality and biological mechanisms and render valid estimates with a minimal bias. This cross-sectional study has not clarified the causality; however, it presents the prevalence of the association between periodontitis and oral health activities, including brushing the lingual area of the teeth. Studies that include biomarker information and demographic factors will further elucidate the biological mechanisms and social impact of the association between periodontitis and oral health activities. In this study, there were no associations between oral health activities, such as fresh fruit consumption or other oral health activities, and periodontal disease, excluding lingual and palatal surface toothbrushing. In addition, there is a possibility that the participants gave false answers during their question-and-answers in order to report “correct” behaviors in the evaluation of lingual toothbrushing, and the Hawthorn effect may have occurred. Additionally, there is a possibility that the results may have been affected by local factors that could not be collected, and it is considered that large-scale group studies are necessary. Also, it was not possible to secure the quality of detailed answers to factors that may affect the outcome evaluation, such as the time for brushing and plaque index, which are the main causal variables. Another obvious limitation of this study is the use of the CPI code to define periodontitis. Based on WHO criteria, periodontitis was defined by three or four points based on the CPI that normally progresses, and classified participants into two groups: non-periodontitis and periodontitis. We included both shallow and deep periodontal pockets in the gingival inflammation group, without classifying subjects according to the degree of gingival inflammation. Future studies that are more representative of the general population with larger sample sizes will permit stratified analysis to detect factors that reduce selection bias and correct the effects of periodontitis on lingual brushing performance. In addition, a study design that considers psychological factors, such as motivation for the patient’s brushing or enhancement of empowerment, will be needed. Despite these limitations, the current study’s results shed insight into the association between periodontitis and lingual site toothbrushing in an elderly cohort of Koreans.

## 5. Conclusions

Although considering the spectrum of physical activity in the elderly, brushing the teeth’s inner surfaces is an important oral health activity in managing periodontal health of the elderly. Therefore, the importance of lingual and palatal site toothbrushing by oneself or others should be considered by the elderly and the caregivers who provide assistance with their oral health care.

## Figures and Tables

**Table 1 ijerph-18-05067-t001:** Distribution of characteristics of participants according to periodontal health status.

Variables		Periodontal Health Status					
		Community Periodontal Index (CPI)			Bleeding on Probing		
		CPI < 3	CPI ≥ 3	*p*-Value	Low	High	*p*-Value
		*n* (%)	*n* (%)		*n* (%)	*n* (%)	
Socio-demographic factors							
Age (years) *		79.3 (7.3)	79.1 (7.0)	0.883	78.3 (7.2)	80.2 (7.1)	0.097
Sex	Male	22 (23.4)	12 (21.4)	0.842	18 (23.4)	16 (21.9)	0.848
	Female	72 (76.6)	44 (78.6)		59 (76.6)	57 (78.1)	
House income	1 quartile	45 (47.9)	37 (66.1)	0.014	35 (45.5)	47 (64.4)	0.007
	2 quartile	28 (29.8)	16 (28.6)		23 (29.9)	21 (28.8)	
	≥3 quartile	21 (22.3)	3 (5.4)		19 (24.7)	5 (6.8)	
Education	≤Elementary school	26 (27.7)	12 (24.4)	0.038	23 (29.9)	15 (20.5)	0.035
	Middle school	34 (36.2)	32 (57.1)		26 (33.8)	40 (54.8)	
	≥High school	34 (36.2)	12 (21.4)		28 (36.4)	18 (24.7)	
Systemic health-related factors							
Activities of daily living	Independent	75 (79.8)	30 (53.6)	0.001	65 (84.4)	40 (54.8)	<0.001
	Dependent	19 (20.2)	26 (46.4)		12 (15.6)	33 (45.2)	
Systemic diseases	Absence	3 (3.2)	2 (3.6)	1	4 (5.2)	1 (1.4)	0.367
	Presence	91 (96.8)	54 (96.4)		73 (94.8)	72 (98.6)	
Smoking	Non-smoker	89 (94.7)	48 (85.7)	0.074	73 (94.8)	64 (87.7)	0.152
	Smoker	5 (5.3)	8 (14.3)		4 (5.2)	9 (12.3)	
Other oral health status							
No. of remaining tooth *		17.6 (7.1)	17.4 (5.8)	0.894	19.9 (6.1)	15.0 (6.3)	<0.001
Oral health activities							
Intake of fresh seasonal vegetables	≥Once a day	87 (92.6)	43 (76.8)	0.011	71 (92.2)	59 (80.8)	0.054
	<Once a day	7 (7.4)	13 (23.2)		6 (7.8)	14 (19.2)	
Need for help in toothbrushing	Independent	90 (95.7)	35 (62.5)	<0.001	74 (96.1)	51 (69.9)	<0.001
	Dependent	4 (4.3)	21 (37.5)		3 (3.9)	22 (30.1)	
Daily frequency of tooth brushing	≥Twice a day	87 (92.6)	45 (80.4)	0.037	72 (93.5)	60 (82.2)	0.044
	<Twice a day	7 (7.4)	11 (19.6)		5 (6.5)	13 (17.8)	
Use of interdental brush	User	40 (42.6)	10 (17.9)	0.002	39 (50.6)	11 (15.1)	<0.001
	Non-user	54 (57.4)	46 (82.1)		38 (49.4)	62 (84.9)	
Brushing of the lingual and palatal surfaces of the teeth	Partially/wholly performed	69 (73.4)	16 (28.6)	<0.001	71 (92.2)	14 (19.2)	<0.001
	Not performed	25 (26.6)	40 (71.4)		6 (7.8)	59 (80.8)	

CPI, community periodontal index. * The values present mean (standard deviation).

**Table 2 ijerph-18-05067-t002:** Association between brushing of inner surfaces of tooth and periodontal health status.

Periodontal Health Status	Brushing of the Inner Surfaces of Teeth	Model 1	Model 2	Model 3	Model 4
		aOR (95% CI)	aOR (95% CI)	aOR (95% CI)	aOR (95% CI)
Periodontitis (CPI ≥ 3)	partially/wholly performed	0.145 (0.064–0.332)	0.148 (0.064–0.345)	0.057 (0.018–0.183)	0.083 (0.024–0.285)
High BOP	partially/wholly performed	0.014 (0.004–0.047)	0.013 (0.004–0.047)	0.011 (0.003–0.047)	0.011 (0.002–0.058)

CPI, community periodontal index; BOP, bleeding on probing; aOR, adjusted odds ratio; CI, confidence interval.

## Data Availability

Original data are available upon request to the corresponding author.

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
