# Peer review of "Effects of Lingual and Palatal Site Toothbrushing on Periodontal Disease in the Elderly: A Cross-Sectional Study"

_ijerph, 2021, doi:10.3390/ijerph18105067_

Round 1

Reviewer 1 Report

This manuscript describes a cross-sectional study analyzes the association between periodontal health status and daily oral health activities including lingual and palatal site tooth 150 Korean participants >65 yr residing in or using care facilities. This manuscript was well written in English and scientifically comprehensive. Clinical data collection and examinations were well described. After correcting for sociological factors,  it was concluded that brushing of the inner surface of teeth was strongly associated with periodontal disease and high bleeding on probing.

Comments to authors:

Table 1. As presented in the downloaded document. Column headers and values are misaligned. Probability values should all be in one column.

Line 17. “…lingual and palatal side brushing.” Should this be “site” brushing to correspond to the title of the manuscript? Throughout the manuscript “side” and not “site” is used in this regards. “Site” is only used in the title. Should the title be revised? The same term should be used throughout.

Lines 170-172. Where is the data to support these statements?

Lines 177-178. “The differences in the characteristics between the high and low BOP…”. Please specify the characteristics to which you are referring.

Lines 235-236. “… with 43.3% of participants partially or wholly not brushing the tooth’s lingual surface (Table 1)”. The categories of brushing the lingual surface in Table 1 are wholly/partial or not performed. The not performed participants are 25 + 40 = 65. So 65/150 is 43.3%. When wholly/partial participants are 69 + 16 = 85. So 85/150 is 56.7%. Please correct sentence.

Lines 251-252. “In addition, the effective size of it among the five activities was the largest.” Effective size of what? Please give a more complete description.

Lines 257-258. “Physical activity was positively associated with antibodies in the orange and blue complex related to healthy periodontal states.” What are antibodies in the orange and blue complex?

Lines 282-283. Grip strength was not measured, but implied from need for help in brushing

Author Response

Review 1)

Table 1. As presented in the downloaded document. Column headers and values are misaligned. Probability values should all be in one column.

 Answer 1)

Based on the review comments, it was confirmed that the table was misaligned in the process of editing, therefore the value and column of Table1 and Table2 was corrected . (Table1-Ln 166, Table2-194)

Review 2)

Line 17. “…lingual and palatal side brushing.” Should this be “site” brushing to correspond to the title of the manuscript? Throughout the manuscript “side” and not “site” is used in this regards. “Site” is only used in the title. Should the title be revised? The same term should be used throughout.

 Answer 2)

Due to the simple notation mistake you mentioned, all terms have been unified and changed “side” to “site”. (Line 18,23,24,29,125,177,186,212,217,224,232,252,337,343)

Review 3)

Lines 170-172. Where is the data to support these statements?

Answer 3)

This notation is a simple classification cut value of main outcome value for the reader's understanding, and the median number of the classification criteria for each major periodontal condition-related variable in Table 1 is separately indicated. In order to avoid confusion for the reader, it does not indicate a separate distribution of descriptive variables other than the classification by major explanatory variables on table. In order to reflect the opinion of the review, to prevent misunderstanding of the relevant part, the contents of Table 1 have been added end of the sentence. (Ln 170-172)

Review 4)

Lines 177-178. “The differences in the characteristics between the high and low BOP…”. Please specify the characteristics to which you are referring.

Answer 4)

Reflecting the review opinion, the ambiguous part was changed to a specific description of the tendency of the group with higher BOP. (Ln 177-180)

Review 5)

Lines 235-236. “… with 43.3% of participants partially or wholly not brushing the tooth’s lingual surface (Table 1)”. The categories of brushing the lingual surface in Table 1 are wholly/partial or not performed. The not performed participants are 25 + 40 = 65. So 65/150 is 43.3%. When wholly/partial participants are 69 + 16 = 85. So 85/150 is 56.7%. Please correct sentence.

Answer 5)

The figures have been revised to reflect the review opinion. I wrote down the percentage of the opposing group and revised it to 56.7% of the group's content according to the opinion you reviewed. (Ln 236-237)

Review 6)

Lines 251-252. “In addition, the effective size of it among the five activities was the largest.” Effective size of what? Please give a more complete description.

Answer 6)

 This part was changed to a more specific description because it was seen as a mistake in selecting words in the part describing the influence of the outcome variable on the main explanatory variable. (Ln 252-256)

Review 7)

Lines 257-258. “Physical activity was positively associated with antibodies in the orange and blue complex related to healthy periodontal states.” What are antibodies in the orange and blue complex?

Answer 7)

The expression has been changed so that the health activity itself creates an anti-body for periodontal disease-related bacteria and maintains a healthy state. (Ln 261-262)

Review 8)

Lines 282-283. Grip strength was not measured, but implied from need for help in brushing

This part was misrepresented while trying to transfer the contents of full-body muscle strength, such as the grip strength summarized in the introduction, and the grip strength part was deleted from the discussion. (Ln 289-290)

Reviewer 2 Report

Thank you very much for letting me review this manuscript of great interest to special need patients.

Was the study approved by an ethics committee?

What kind of toothbrush was recommended to patients?

Agree that the socio-economic status affects the overall health of the individual, but a correct motivation allows to guarantee a minimum resolution of the periodontal indices.

Allow me to suggest a quote to add in such a way that the message can be conveyed that the use of a sonic or roto-oscillating toothbrush can further reduce the plaque and bleeding indices than a manual toothbrush, and I would insert the part of instructions for using an electric toothbrush to caregivers. 

The efficacy of powered oscillating heads vs. Powered sonic action heads toothbrushes to maintain periodontal and peri-implant health: A narrative review
International Journal of Environmental Research and Public Health, 2021, 18(4), pp. 1–17, 1468

Author Response

Review 1)

Was the study approved by an ethics committee?

Answer 1)

Thanks for the good review comment. In order to convey the research ethics review of the thesis to the reader, the IRB review contents and the review number are indicated in the IRB statements part. (Ln 357-359)

Review 2)

What kind of toothbrush was recommended to patients?

Answer 2)

As a result of the research, in order to make it easier to brush the lingual and palatal surface partially or entirely, a toothbrush that has a smaller head size and can effectively brush between teeth is required to be recommended. In addition, it has been reinforced that a recommendation for an electric toothbrush that can brush more effectively is required when brushing by a caregiver is provided. Therefore, the content was added to the conclusion by reflecting the review opinion. (Ln 274-277)

Review 3)
Agree that the socio-economic status affects the overall health of the individual, but a correct motivation allows to guarantee a minimum resolution of the periodontal indices.

Answer 3)

As given in the review opinion, various factors related to health behavior other than socioeconomic factors were adjusted, and factors such as motivation and empowerment of each participant will be reinforced in future studies. These contents have been reinforced in the limit section.(Ln 330-332)

Reivew 4)
Allow me to suggest a quote to add in such a way that the message can be conveyed that the use of a sonic or roto-oscillating toothbrush can further reduce the plaque and bleeding indices than a manual toothbrush, and I would insert the part of instructions for using an electric toothbrush to caregivers. 

The efficacy of powered oscillating heads vs. Powered sonic action heads toothbrushes to maintain periodontal and peri-implant health: A narrative review

International Journal of Environmental Research and Public Health, 2021, 18(4), pp. 1–17, 1468

Answer 4)

To reflect this opinion, we have added the requirements for toothbrushes and a review of the thesis on electric toothbrushes on Reference No. 40 (Ln 275-277)

Reviewer 3 Report

In the study entitled “Effects of lingual and palatal site toothbrushing on periodontal disease in the elderly: a cross-sectional study“, the authors tried to analyze the association between periodontal health status and daily oral health activities including lingual and palatal side toothbrushing. Clinical examination regarding oral health status, including periodontal health status was evaluated, and data on the oral health activities, socio-demographic factors, and systemic health-related factors were obtained using a questionnaire. Statistical analyses assessed the differences of periodontal health status according to daily oral health activities, including lingual and palatal side toothbrushing.

The article is described in detail in accordance with the objectives in a methodical and result-oriented manner. From this point of view, there are no objections on the part of the reviewer. In the discussion, however, the reviewer would have added three more elements:

  • The split-mouth design
  • Hawthorne effect
  • Local factors

The split-mouth design

The split-mouth design has been one of the main investigation methods used in periodontal clinical studies for many years (Hujoel and Loesche, 1990). The split-mouth division into two experimental units shows the greatest scientific efficiency compared to other divisions and should be therefore chosen in this study. The split-mouth design that should be used should divide the dentition into right and left halves, following Ramfjord et al. (1968). By combining the split-mouth design with the cross-over design in the reviewed study, the investigations could be performed with a significantly reduced number of patients (Elbourne et al., 2002). The possibility of a residual effect with a change in application can lead to confounding of the results in a study with a cross-over design (Elbourne et al., 2002). The risk that the results in the reviewed study were influenced by a residual effect will be significantly reduced by a washout period between application intervals of three months. The choice of the cross-over design allowed each patient to experience both therapies, i.e., normal oral hygiene and normal oral hygiene with additive therapeutic use, and to report their subjective feelings to the investigator in the mentioned questionnaire at the end of the study. According to Antczak-Bouckoms et al. (1990), the cross-over design should only be performed in patients with chronic diseases and a relatively stable oral situation. Both of these were true for the participants in this study with chronic periodontitis on maintenance therapy.

Hawthorne effect

As is generally the case in connection with oral hygiene studies, it is conceivable in this study that the Hawthorne effect had an influence on the results obtained (Campbell et al., 1995). The assumption here is that the very act of participating in a study causes participants in a study to change their natural behavior because they know they are participating in a study and are under observation. Consequently, it is possible that the subjects in this study improved their normal oral hygiene behavior for the period of the study by taking more time for more thorough and better dental care.

The current body of research suggests that women are more concerned with their oral hygiene behaviors and more interested in maintaining their oral health than men (Ostberg et al., 1999; Kateeb, 2010). Despite the evidence of improved and intensified oral hygiene among women, men often exhibit better oral situations when compared by age (Zubiene et al., 2009).

However, it cannot be ruled out that interindividual differences in effectiveness due to differing motivation or skill of the subjects were present, which is generally considered to be a limiting factor in studies evaluating the effectiveness of oral hygiene interventions.

Local factors

Local factors such as defective dental restorations, which may negatively influence the effectiveness of personal oral hygiene and thus the possibility of successful removal of bacterial plaque, were not assessed in the study patients. For this reason, among others, the study participants represented the typical characteristics of patients in maintenance therapy as required by O'Mullane et al. (2012). It should be noted that it is certainly more difficult to record a positive effect on the test side, the better the oral hygiene condition is to the baseline finding and on the control side, respectively.

Author Response

In the study entitled “Effects of lingual and palatal site toothbrushing on periodontal disease in the elderly: a cross-sectional study“, the authors tried to analyze the association between periodontal health status and daily oral health activities including lingual and palatal side toothbrushing. Clinical examination regarding oral health status, including periodontal health status was evaluated, and data on the oral health activities, socio-demographic factors, and systemic health-related factors were obtained using a questionnaire. Statistical analyses assessed the differences of periodontal health status according to daily oral health activities, including lingual and palatal side toothbrushing.

The article is described in detail in accordance with the objectives in a methodical and result-oriented manner. From this point of view, there are no objections on the part of the reviewer. In the discussion, however, the reviewer would have added three more elements:

  • The split-mouth design
  • Hawthorne effect
  • Local factors

Answer 1)

To describe the limitations of the study and the necessity of further research, a description of the need for split mouth design in future clinical studies (Ln 304-307), the possibility of the Hawthorne effect according to the questionnaire (Ln 315-318), and the possibility of fluctuations in the results due to local factors not measured were added to the limitation part in the limitation part (Ln 318-320).

Reviewer 4 Report

The authors aimed to explore oral hygiene behaviors (instructions) that should  be focused on for plaque control, which is the first step in preventing periodontal inflammation or periodontal diseases, as evaluated by the association between periodontal  health status and oral health activities. This cross-sectional study demonstrated that the lingual and palatal side toothbrushing  was associated with periodontitis in elderly Koreans. The proposed results would provide baseline data to direct oral health education  for the elderly and their caregivers

The study is easy to follow and covers an interesting topic, but some issues should be improved before publication. The manuscript needs moderate English change and grammar correction. Please also check typos thorough the text.

Introduction section: Will be very useful for the readers, as also stated from the authors, to stress better the emerging role of oral microbial dysbiosis  in the progression of the disease  (please see and discuss: PMID: 32560235 ; AND    PMID: 32397555).

Conclusion Section: This paragraph required a general revision to improve the same, and to add some "take-home message".

Author Response

The authors aimed to explore oral hygiene behaviors (instructions) that should  be focused on for plaque control, which is the first step in preventing periodontal inflammation or periodontal diseases, as evaluated by the association between periodontal  health status and oral health activities. This cross-sectional study demonstrated that the lingual and palatal side toothbrushing  was associated with periodontitis in elderly Koreans. The proposed results would provide baseline data to direct oral health education  for the elderly and their caregivers

The study is easy to follow and covers an interesting topic, but some issues should be improved before publication. The manuscript needs moderate English change and grammar correction. Please also check typos thorough the text.

Introduction section: Will be very useful for the readers, as also stated from the authors, to stress better the emerging role of oral microbial dysbiosis  in the progression of the disease  (please see and discuss: PMID: 32560235 ; AND    PMID: 32397555).

Conclusion Section: This paragraph required a general revision to improve the same, and to add some "take-home message".

Answer 1)

In reflection of the review opinion, the emphasis on the role of oral microbial dysbiosis in periodontal disease was reinforced in the introduction section. (Ln 52-54)

In the conclusion section, we have reinforced the contents related to the recommended toothbrush shape to convey the related message at home. (Ln 273-277)